# Alternating Lenvatinib and Trans-Arterial Therapy Prolongs Overall Survival in Patients with Inter-Mediate Stage HepatoCellular Carcinoma: A Propensity Score Matching Study

**DOI:** 10.3390/cancers13010160

**Published:** 2021-01-05

**Authors:** Shigeo Shimose, Hideki Iwamoto, Masatoshi Tanaka, Takashi Niizeki, Tomotake Shirono, Yu Noda, Naoki Kamachi, Shusuke Okamura, Masahito Nakano, Hideya Suga, Taizo Yamaguchi, Takumi Kawaguchi, Ryoko Kuromatsu, Kazunori Noguchi, Hironori Koga, Takuji Torimura

**Affiliations:** 1Division of Gastroenterology, Department of Medicine, Kurume University School of Medicine, Kurume 830-0011, Japan; niizeki_takashi@kurume-u.ac.jp (T.N.); shirono_tomotake@med.kurume-u.ac.jp (T.S.); noda_yuu@med.kurume-u.ac.jp (Y.N.); kamachi_naoki@med.kurume-u.ac.jp (N.K.); okamura_shyuusuke@kurume-u.ac.jp (S.O.); nakano_masahito@kurume-u.ac.jp (M.N.); takumi@med.kurume-u.ac.jp (T.K.); ryoko@med.kurme-u.ac.jp (R.K.); hirokoga@med.kurume-u.ac.jp (H.K.); tori@med.kurume-u.ac.jp (T.T.); 2Department of Gastroenterology and Hepatology, Iwamoto Internal Medical Clinic, Kitakyusyu 802-0832, Japan; iwamotos@orion.ocn.ne.jp; 3Department of Gastroenterology and Hepatology, Yokokura Hospital, Miyama, Fukuoka 839-0295, Japan; mark7@yokokura-hp.or.jp; 4Department of Gastroenterology and Hepatology, Yanagawa Hospital, Fukuoka 832-0077, Japan; suga516@med.kurume-u.ac.jp; 5Department of Gastroenterology and Hepatology, Omuta City Hospital, Fukuoka 836-8567, Japan; hisyo@ghp.omuta.fukuoka.jp

**Keywords:** lenvatinib, alternating therapy, intermediate stage, TACE, HAIC

## Abstract

**Simple Summary:**

This study aimed to investigate the efficacy of alternating lenvatinib (LEN) and trans-arterial therapy (AT) in patients with intermediate-stage hepatocellular carcinoma (HCC) after propensity score matching analysis. AT and albumin-bilirubin (ALBI) grade 1 were identified as independent factors for overall survival in patients with intermediate-stage HCC. Decision tree analysis demonstrated that the recommended indication of AT was below 70 years of age with ALBI grade 1. This study may reveal clinical features associated with the efficacy of AT and may contribute to improving survival in patients with intermediate-stage HCC.

**Abstract:**

We aimed to evaluate the impact of alternating lenvatinib (LEN) and trans-arterial therapy (AT) in patients with intermediate-stage hepatocellular carcinoma (HCC) after propensity score matching (PSM). This retrospective study enrolled 113 patients with intermediate-stage HCC treated LEN. Patients were classified into the AT (*n* = 41) or non-AT group (*n* = 72) according to the post LEN treatment. Overall survival (OS) was calculated using the Kaplan–Meier method and analyzed using a log-rank test after PSM. Factors associated with AT were evaluated using a decision tree analysis. After PSM, there were no significant differences in age, sex, etiology, or albumin-bilirubin (ALBI) score/grade between groups. The survival rate of the AT group was significantly higher than that of the non-AT group (median survival time; not reached vs. 16.3 months, *P* = 0.01). Independent factors associated with OS were AT and ALBI grade 1 in the Cox regression analysis. In the decision tree analysis, age and ALBI were the first and second splitting variables for AT. In this study, we show that AT may improve prognosis in patients with intermediate-stage HCC. Moreover, alternating LEN and trans-arterial therapy may be recommended for patients below 70 years of age with ALBI grade 1.

## 1. Introduction

Liver cancer is the third leading cause of cancer-related deaths and the sixth most common neoplasm [1,2]. Hepatocellular carcinoma (HCC) is the most common primary liver cancer. Although the prognosis of patients with early-stage HCC has been improved by the development of curative therapies [3,4], patients with advanced HCC are usually treated with transcatheter arterial chemoembolization (TACE) [5,6], hepatic arterial infusion chemotherapy (HAIC) [7], molecular-targeted agents (MTAs) [3], and immunotherapy [8] and the prognosis remains poor.

The Barcelona Clinic Liver Cancer staging system is widely used for liver cancer classification. In this system, the intermediate stage is quite board and includes a heterogeneous patient population. Although only TACE is recommended for intermediate-stage HCC patients [9,10], MTAs have also become available for intermediate-stage HCC patients with preserved liver function [11]. Recent studies have shown that TACE plus sorafenib (SORA) significantly improves progression-free survival (PFS) over treatment with TACE alone in patients with unresectable HCC [12]. Moreover, SORA plus HAIC improves overall survival (OS) compared to SORA in patients with HCC [7].

Lenvatinib (LEN) is one of the MTAs approved for first-line treatment of patients with unresectable advanced HCC in the USA, European Union, Japan, and China, based on the results of the REFLECT trial, a global multicenter randomized phase 3 trial of LEN for HCC [13]. Kudo et al. had previously reported that LEN is associated with better OS than TACE in patients with intermediate-stage HCC beyond the up-to-seven criteria in A Proof-Of-Concept Study [14]. However, it is also associated with a high rate of adverse events (AEs), which leads to discontinuation of LEN treatment and poor prognosis in patients [15]. Thus, the how to use and continuous administration period of LEN in patients with unresectable HCC is an important issue.

Advanced HCC usually consists of heterogeneous nodules. We have frequently encountered new nodules whose growth could not be suppressed even though LEN treatment controlled the tumor growth of other nodules. In such cases, LEN treatment is substituted with sequential MTAs [16]. However, if new, uncontrolled nodules could be treated with TACE or HAIC, it might be possible to continue LEN treatment—that has a higher response rate compared to other MTAs—and improve the prognosis of patients with intermediate-stage HCC.

In this study, we aimed to identify the therapeutic efficacy of alternating LEN and trans-arterial therapy (AT), including TACE and HAIC, in patients with intermediate-stage HCC. In addition, we applied propensity score matching analysis (PSM) to reduce confounding.

## 2. Results

### 2.1. Patient Characteristics before PSM

Patient profiles are summarized in Table 1. The median age was 75 (42–90 years), and 17.2% (21/113) of patients were female. The etiology of liver diseases comprised non-hepatitis B or C virus in 35.3% (40/113) of patients, the median albumin-bilirubin (ALBI) score was −2.4, and ALBI grade 1 was observed in 38.9% (44/113) of patients (Table 1). The median tumor size was 31 mm, and 73.4% of patients had ≥ 5 tumors. TACE condition before treatment with LEN was refractory TACE in 84.0% (95/113) of patients, and the rest were ineligible for TACE [11]. Median alpha-fetoprotein (AFP) and des-gamma-carboxy prothrombin (DCP) levels were 25.5 ng/mL (1.5–209,018 ng/mL) and 101.5 mAU/mL (11.5–179,531 mAU/mL), respectively. Patients who received the AT regime comprised 36.2% (41/113) of total patients, and the number of patients treated with TACE, HAIC, and TACE+HAIC was 26, 8, and 7, respectively. The median age, ALBI score, and ALBI grade were significantly higher in the AT group than in the non-AT group; however, there were no significant differences in sex, cause of HCC, tumor size, tumor number, TACE condition before treatment with LEN, or AFP and DCP levels in the two groups (Table 1).

### 2.2. Evaluation with mRECIST after Initial Treatment with LEN

Evaluated Initial response to LEN after four–six weeks, overall complete response, partial response, stable disease, and progression disease were observed in 5.3% (6/113), 39.8% (45/113), 40.7% (46/113), and 14.2% (16/113) of patients, respectively (Figure 1). The overall objective response rate (ORR) and disease control rate (DCR) were 45.1% (51/113) and 85.8% (97/113), respectively. The initial antitumor effect of LEN is schematized as a waterfall plot (Figure 1).

### 2.3. Swimmer Plot of Assessment by Investigator Assessment

The swimmer plot of patients treated with AT is shown in Figure 2. Of the 41 patients treated with AT, 51% (21/41) continued AT treatment until the cut-off date, and 63.4, 19.5, and 17.1% were treated with TACE, HAIC, and TACE+HAIC, respectively. The median medication period with AT was 13.6 months (Figure 2).

### 2.4. Kaplan–Meier Curves for OS before PSM

Overall survival in the AT group was significantly higher than that in the non-AT group (*P* < 0.001, Figure 3A). The 1- and 2-year survival rates for the AT group were 88 and 79%, and those for the non-AT group were 67 and 38%, respectively (Figure 3A).

### 2.5. Patient Characteristics after PSM

To minimize the effect of confounding factors, we performed PSM analysis using the following factors: age, sex, etiology, ALBI score, tumor size, tumor number, AFP value, and DCP value. There was no significant difference between the AT and non-AT groups in any variable (all *P* > 0.05, Table 2).

### 2.6. Kaplan–Meier Curves for OS after PSM

Kaplan–Meier curves for OS were evaluated after PSM. The OS period in the AT group was significantly longer than that in the non-AT group (*P* = 0.01, Figure 3B). The 1- and 2- year survival rates in the AT group were 83 and 66%, and those in the non-AT group were 71 and 28%, respectively (Figure 3B).

### 2.7. Effect of AT Intervention on LEN Administration Period

To clarify whether intervention with AT contributed to prolongation of the LEN treatment, we compared the duration of treatment with LEN in the AT and non-AT groups after PSM. The administration period of LEN in the AT and non-AT groups was 13.7 and 8.6 months, respectively. Moreover, there was a significant difference in OS between the two groups (*P* = 0.041) (Appendix A).

### 2.8. Univariate and Multivariate Analyses of Factors Associated with OS after PSM

ALBI grade 1 (*P* = 0.025) and AT (*P* = 0.014) were selected as variables via univariate analysis. In the multivariate analysis, ALBI grade 1 (*P* = 0.011) and AT (*P* = 0.009) were identified as independent factors for OS (Table 3).

### 2.9. Decision Tree Analysis for AT

In this study, the AT rate in all subjects was 36% at the time of study cessation. To determine the profiles associated with AT, a decision tree analysis was performed which revealed that age was the first splitting variable for AT rate. In patients aged < 70 years, the ALBI grade was determined to be the second split, with an AT rate of 66% in patients with ALBI grade 1 (Figure 4). Although the AT rate was 25% in patients aged ≥ 70 years, it was 41% in those with ALBI grade 1 (Figure 4).

### 2.10. Severe Adverse Events from the Treatment (Grade ≥ 3)

Adverse events—as determined by the attending physician—are shown in Appendix A. In the AT group, hypertension and proteinuria occurred in 17% (7/41) and 12.2% (5/41) of the patients, respectively. Similarly, in the non-AT group, hypertension and proteinuria occurred in 16.7% (12/72) and 15.2% (11/72) of the patients, respectively (Appendix A). There were no significant differences between the two groups in any parameter (all *P* > 0.05, Appendix A). On the other hands, there were no biliary complications due to trans-arterial therapy including TACE and HAIC in the study.

## 3. Discussion

Although TACE is the standard treatment for intermediate-stage HCC [9], it is ineffective for suppressing tumor growth in multiple HCC nodules (≥7) when administered alone. LEN is the only first-line agent to demonstrate a survival benefit over TACE in TACE-naive patients out of up-to-seven criteria [14]. In cases where new nodules exhibiting treatment resistance to LEN are detected, a change from LEN to second-line MTAs is recommended.

In this study, HCC patients with TACE refractoriness or those ineligible for TACE received prolonged treatment with LEN for other HCC nodules. This was likely due to additional treatment with TACE or HAIC for tumor nodules that were not controlled by LEN monotherapy. We also demonstrated a beneficial impact of alternating LEN and trans-arterial therapy compared to LEN monotherapy on the prognosis of patients with intermediate-stage HCC after PSM. In multivariate analysis after PSM, AT therapy and ALBI grade 1 were identified as independent, reliable prognostic factors in patients with intermediate HCC. Hiraoka et al. reported that ALBI grade is useful for assessing hepatic function and HCC prognosis [17]. ALBI grade 1 and preserved liver function are associated with OS or response rate in LEN treatment [14]. Moreover, Lee et al. reported that ALBI grade is an important factor associated with survival in patients with intermediate-stage HCC who underwent TACE [18]. In our study, 36.6% of patients received HAIC treatment along with alternating therapy with LEN. Recently, Wang X et al. reported that the prognosis of patients treated SORA plus HAIC prolonged OS compared treated with SORA monotherapy for advanced HCC [19]. Moreover, Shi et al. reported that HAIC is suitable for large unresectable HCC, such as high tumor burden (multinodular type, huge type, and invasive growth type) [20]. These therapies are also characterized by a high rate of locally controlled tumor growth. They used FOLFOX regimen (5-fluorouracil, oxaliplatin and folinic acid) as a HAIC regimen, but we used cisplatin with lipiodol plus 5-FU regimen. Although there is a difference in a HAIC regimen between their reports and the present study, these findings support that long-term treatment with AT can prolong the OS period compared to LEN monotherapy in patients with preserved liver function in intermediate-stage HCC.

LEN suppresses angiogenesis by suppressing vascular endothelial growth factor (VEGF) receptors 1–3, fibroblast growth factor (FGF) receptors 1–4, platelet-derived growth factor receptor α, RET, and KIT [21,22]. However, TACE or HAIC induce ischemic conditions in tumor tissues, which upregulates hypoxia-inducible factor 1-α expression [22,23], leading to increased production of VEGF, FGF, HGF, and other angiogenic factors in tumor tissues [24]. High levels of angiogenic factors stimulate tumor recurrence or growth. Yang et al. have previously reported that discontinuation of anti-VEGF therapy promotes metastasis through a liver revascularization mechanism [25]. Thus, LEN administration after TACE or HAIC might have suppressed the effects of angiogenic factors and tumor recurrence in our study.

In the REFLECT study, the ORR and DCR of LEN treatment were superior to those of SORA treatment. However, LEN was not shown to be superior to SORA as a primary treatment in terms of OS duration in patients with unresectable, untreated HCC [13]. In our previous study using PSM, despite a higher response rate, primary treatment with LEN was not confirmed to be superior to SORA in terms of OS in patients with unresectable advanced HCC [26]. In addition, LEN was associated with a significantly higher AE rate and lower transition rate to second-line MTAs compared to SORA [26]. A decision tree analysis revealed that alternating LEN and trans-arterial therapy was recommended for intermediate-stage HCC patients below 70 years old with ALBI grade 1. LEN has been reported to have some significant AEs [27]. We have previously reported that advanced age is associated with the discontinuation of LEN due to severe AEs [15]. It is possible that receiving trans-arterial therapy immediately after LEN treatment might be a severe physical load for patients. These are vulnerable to over-treatment (high likelihood of complications/toxicity) [28]. Accordingly, younger age may be associated with the variable initial split for AT.

The present study has several limitations. First, the study design was a retrospective study, and the sample size was small. Second, there was a selection bias for the classification of the AT and non-AT groups. Third, the non-AT group was heterogeneous and was not an appropriate control group to the AT group. Fourth, we did not evaluate the history of previous treatment and post-treatment. Thus, a randomized, controlled, and prospective validation study with a larger number of intermediate-stage HCC patients is required to determine the efficacy of AT.

## 4. Materials and Methods 

### 4.1. Study Design and Patients

This retrospective study evaluated 206 patients with unresectable HCC who were treated with LEN between 24 March 2018 and 31 July 2020 across the following 5 institutions: Kurume University Hospital (Kurume, Japan), Yokokura Hospital (Miyama, Japan), Iwamoto Internal Medical Clinic (Kitakyushu, Japan), Omuta City Hospital (Omuta, Japan), and Yanagawa Hospital (Yanagawa, Japan). The cut-off date for this analysis was 31 August 2020. Following the initial evaluation, 93 patients with the following exclusion criteria were excluded from analysis: early stage, 4 patients; advanced stage, 87 patients; insufficient treatment time, 2 patients (Appendix A). A total of 113 patients were enrolled and classified as AT (*n* = 41) or non-AT (*n* = 72) (Appendix A). After PSM, treatment efficacy was compared between 24 AT-treated patients and 24 non-AT-treated patients (Appendix A). An opt-out approach was used to obtain informed consent from the patients and personal information was protected during data collection. This protocol conformed to the rules of the Declaration of Helsinki and received approval from the ethics committee of Kurume University (approval number: 20192).

### 4.2. PSM Analysis

PSM overcomes different distributions of covariates among individuals allocated to specific interventions and was generated using potential covariates that could affect group allocation [29]. In this study, propensity scores for all patients were estimated by a logistic regression model using the following baseline characteristics as covariates: age, sex, etiology, tumor size, tumor number, ALBI score [30], and AFP and DCP levels. A one-to-one nearest-neighbor matching algorithm with an optimal caliper of 0.2 without replacement was used to generate 48 pairs of patients. Since P-values could be biased by population size, the propensity score matching results were also reported as effect sizes: |value| < 0.2 indicated a negligible difference, |value| < 0.5 indicated a small difference, |value| < 0.8 indicated a moderate difference, and any other value indicated a large difference. The c-statistic was 0.80 (Appendix A).

### 4.3. LEN Treatment Protocol

LEN (Eisai Co., Ltd, Tokyo, Japan) was administered to patients with unresectable HCC. The standard dose of LEN therapy was determined based on body weight and liver function according to the manufacturers’ instructions. LEN was orally administered at a dose of 12 mg/day for patients with body weight ≥ 60 kg or 8 mg/day for patients with body weight < 60 kg. According to the guidelines for the administration of LEN, the drug dose was reduced or the treatment was interrupted in patients who developed severe adverse events (grade ≥ 3).

### 4.4. Evaluation of the Therapeutic Response and Follow-Up Schedule

Therapeutic response was evaluated using dynamic computed tomography or magnetic resonance imaging at 4–6 weeks after the initiation of treatment with LEN according to the Modified Response Evaluation Criteria in Solid Tumors (mRECIST) system [31], and thereafter at intervals of 2–3 months until death or study cessation.

### 4.5. Alternating LEN+ Trans-Arterial Therapy

In cases where a therapeutic response to LEN treatment was detected, we continued the treatment. However, for cases that were diagnosed with disease progression, where tumor vascularity resumed after having disappeared previously, or where a new lesion appeared, we administered the recommended AT (LEN+ trans-arterial therapy) to patients with tolerance to LEN. LEN was discontinued 2 days before the trans-arterial therapy was administered. Within 2 weeks after the trans-arterial chemotherapy, LEN was restarted at the same dose or half the dose, depending on the patient’s condition. Although we first administered TACE as an additional treatment according to the evidence-based clinical practice guidelines for intermediate-stage HCC [9], HAIC was administered in patients who were considered unsuitable for TACE—such as those with multinodular or invasive growth types [11].

### 4.6. TACE Treatment Protocol

Angiography was performed for the celiac artery and the common hepatic artery using a 3- or 4-Fr catheter, and digital subtraction angiography was performed with a nonionic iodine contrast agent. The tumor-containing segment was evaluated using imaging techniques including cone-beam computed tomography. Subsequently, a 1.7- or 1.9-Fr microcatheter (Piolax Inc., Kanagawa, Japan) was inserted into the sub- or sub-sub-hepatic segment to locate the tumor using the adapted microwire (Piolax Inc.). The catheter was advanced toward the tumor-feeding artery. Conventional TACE was performed using 20–50 mg of epirubicin (Nippon Kaasku Co., Ltd., Tokyo, Japan) or cisplatin (Nippon Kayaku Co., Ltd) with lipiodol (Guerbet Co., Ltd., Tokyo, Japan), depending on the size and number of tumors, and was absorbable gelatin sponge particles (Nippon Kayaku Co., Ltd.) [5]

### 4.7. HAIC Treatment Protocol

HAIC was conducted after the insertion of an implanted catheter (Piolax Inc.) An indwelling catheter (5-Fr W-spiral Catheter; Piolax) was inserted through the right femoral artery, with the distal end of the catheter extending into the hepatic artery or gastroduodenal artery, and the proximal end connected to the port system (SOPH-A-PORT; Sophysa, Besançon, France). Following the inpatient regimen of HAIC, 50 mg of fine-powder cisplatin was suspended in 5–10 mL of lipiodol, for which the suspension volume was decided by tumor volume. On day 1, a cisplatin-lipiodol suspension was injected from the implanted catheter under angiography, followed by the injection 250 mg of 5-FU. Then, 1250 mg of 5-FU was continuously infused for 5 days using an infusion balloon pump (SUREFUSER PUMP, Nipro Pharma Corporation, Osaka, Japan). This regimen was administered once a week for the first 2 weeks [32].

### 4.8. Safety Evaluation

Safety was continuously evaluated via monitoring of vital signs and clinical laboratory test results and assessment of the incidence and severity of adverse events (AEs) according to the National Cancer Institute Common Terminology Criteria for Adverse Events (CTCAE), version 4.0. In this study, we also recorded AEs after the trans-arterial therapy treatment. AEs were defined as those classified as grade ≥ 3 according to the CTCAE.

### 4.9. Statistical Analysis

All data are presented as either the number or the median (range). All statistical analyses were performed using a statistical analysis software (JMP Pro version 14, SAS Institute Inc., Cary, NC, USA). OS was calculated using the Kaplan–Meier method and analyzed using the log-rank test. We also performed a decision tree analysis to identify factors associated with the possibility of administering AT, as described previously [29]. The Swimmer plot was used to describe the administration history and tumor responses based on investigator assessment. Univariate and multivariate analyses were conducted using the Cox proportional hazards model to identify risk factors associated with OS. A two-tailed *P*-value of < 0.05 was considered statistically significant.

## 5. Conclusions

In conclusion, we demonstrated that after PSM, AT with LEN and trans-arterial therapy improved prognosis compared to LEN monotherapy in patients with intermediate-stage HCC. Moreover, the decision tree analysis revealed that the AT regime may be recommended for intermediate-stage HCC patients below 70 years old who have ALBI grade 1.

## Figures and Tables

**Figure 1 cancers-13-00160-f001:**
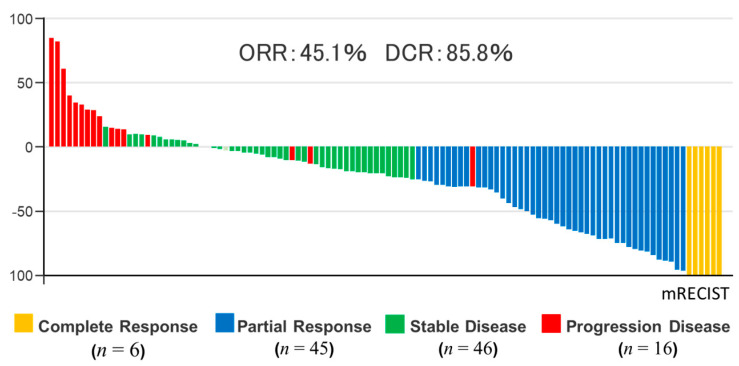
Waterfall plot showing maximum changes in tumor size in patients treated with lenvatinib (LEN). The waterfall plot was performed by the initial response to LEN evaluated. Target regions of tumors were examined according to the Modified Response Evaluation Criteria in Solid Tumors (mRECIST) system.

**Figure 2 cancers-13-00160-f002:**
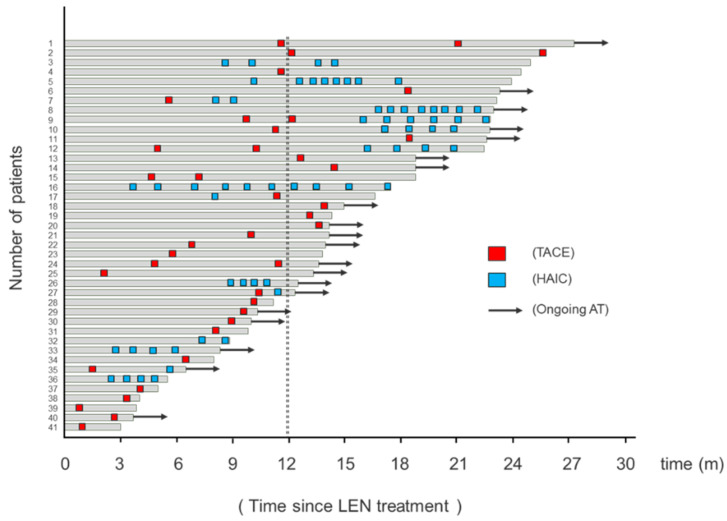
Swimmer plot of assessment by investigator assessment with alternating LEN and trans-arterial therapy (AT). Red squares indicate trans-arterial chemoembolization (TACE) treatment, and blue squares indicate hepatic arterial infusion chemotherapy (HAIC). Arrows indicate ongoing AT. The black dotted line indicates 12 months from the beginning of lenvatinib (LEN) administration.

**Figure 3 cancers-13-00160-f003:**
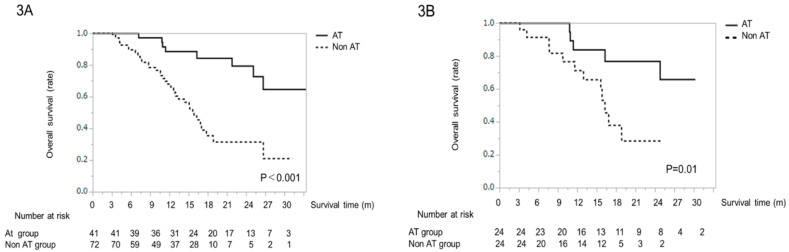
(**A**) Overall survival time in patients treated with lenvatinib (LEN), before propensity matching score (PSM) analysis. Kaplan–Meier curves of overall survival for the alternating LEN and trans-arterial therapy (AT, solid line) and non-AT (dotted line) groups. (**B**) Overall survival time in patients treated with LEN, after PSM analysis. Kaplan–Meier curves for overall survival for the AT (solid line) and non-AT (dotted line) groups.

**Figure 4 cancers-13-00160-f004:**
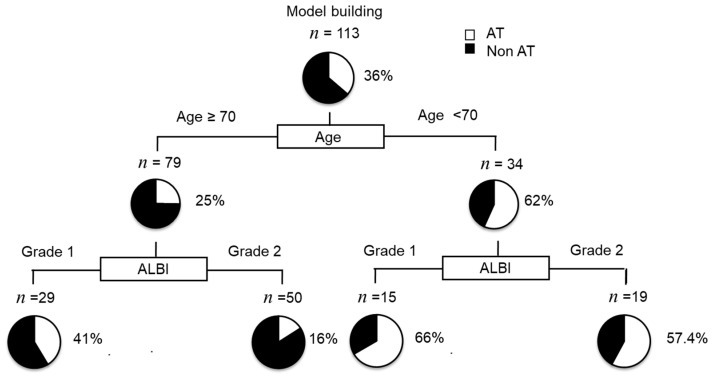
Profiles associated with alternating lenvatinib (LEN) and trans-arterial therapy (AT) in hepatocellular carcinoma (HCC) patients treated with LEN, and the decision tree algorithm for AT. The pie graphs indicate the percentage of AT (white)/non-AT (black) in each group.

**Table 1 cancers-13-00160-t001:** Patient characteristics.

Characteristic	All Patients	Alternating Therapy	No Alternating Therapy	*p*
*n*	113	41	72	
Age (years old)	75 (42–90)	69 (42–83)	78 (60–90)	<0.001
Sex (female/male)	21/92	7/34	14/58	0.755
Etiology (HBV/HCV/Others)	16/57/40	9/15/17	7/42/23	0.053
ALBI score (Median (rage))	−2.40 (−3.60–−1.53)	–2.67 (−3.60–−1.70)	−2.29 (−3.12–−1.53)	<0.001
ALBI grade (1/2/3)	44/69/0	22/19/0	22/50/0	<0.001
Number tumors <5/≥5	30/83	10/31	20/52	0.695
tumor size (mm)	31 (10–170)	31 (11–170)	31 (10–114)	0.806
Up-to-seven criteria (within/out)	16/97	6/35	10/62	0.913
Diabetes mellitus (+/−)	56/57	24/17	32/40	0.149
TACE condition before LEN treatment (refractory/not eligible)	95/18	32/9	63/9	0.193
AFP (ng/mL)	25.5 (1.5–209,018)	19.8 (1.9–60,483)	32.5 (1.9–209,018)	0.208
DCP (mAU/mL)	101.5 (11.5–179,531)	137 (14–57,304)	79 (11.5–179,531)	0.436

Note. Data are expressed as median (range), or number. Abbreviations: HCC, hepatocellular carcinoma; HBV, hepatitis B virus; HCV, hepatis C virus; ALBI score, Albumin-bilirubin score; TACE, transcatheter arterial chemoembolization; LEN, lenvatinib; AFP, α-fetoprotein; DCP, des-γ-carboxy prothrombin.

**Table 2 cancers-13-00160-t002:** Patient characteristics after propensity score matching.

Characteristic	All Patients	Alternating Therapy	No Alternating Therapy	*p*
*n*	48	24	24	
Age (years old)	72 (42–84)	75 (42–83)	71 (60–84)	0.661
Sex (female/male)	11/37	6/18	5/19	0.731
Etiology (HBV/HCV/Others)	7/22/19	2/11/11	5/11/8	0.414
ALBI score (Median (rage))	−2.52 (−3.28–−1.56)	−2.56 (−3.28–−1.70)	−2.48 (−3.07–−1.57)	0.358
ALBI grade (1/2/3)	19/29/0	11/13/0	8/16/0	0.378
Number tumors <5/≥5	10/38	6/18	4/20	0.477
Tumor size (mm)	30.5 (11–158)	28 (11–120)	28 (11–87)	0.909
Up-to-seven criteria (within/out)	5/43	3/21	2/22	0.637
Diabetes mellitus (+/-)	23/25	12/12	11/13	0.772
AFP (ng/mL)	14 (1.6–60,483)	9.4 (2.1–60,483)	18.1 (1.6–6800)	0.451
DCP (mAU/mL)	115 (14–57,304)	120 (14–57,304)	14 (14–6200)	0.909

Note. Data are expressed as median (range), or number. Abbreviations: LEN, lenvatinib; HCC, hepatocellular carcinoma; HBV, hepatitis B virus; HCV, hepatis C virus; ALBI score, Albumin-bilirubin score; AFP, α-fetoprotein; DCP, des-γ-carboxy prothrombin.

**Table 3 cancers-13-00160-t003:** Univariate and multivariate analyses of factors associated with OS after propensity score matching.

Variable	Univariate Analysis	Multivariate Analysis		
	*P*-Value	Odds Ratio	95% CI	*P*-Value
Age, <70 vs. ≥70	0.374			
Sex, female vs. male	0.612			
Cause of HCC, HBV vs. HCV vs. other	0.822			
ALBI grade, 1 vs. 2	0.025	0.249	0.072–0.716	0.011
Number tumors, <5 vs. ≥5	0.955			
Tumor size <30 vs. ≥30	0.717			
Up-to-seven criteria, within vs. beyond	0.977			
Diabetes mellitus, (presence vs. absence)	0.876			
Alternating therapy (+/−)	0.014	0.239	0.054–0.745	0.009
AFP, <100 vs. ≥100 ng/mL	0.325			
DCP, <100 vs. ≥100 mAU/mL	0.232			

Note. Data are expressed as median (range), or number. Abbreviations: HCC, hepatocellular carcinoma; HBV, hepatitis B virus; HCV, hepatis C virus; ALBI score, Albumin-bilirubin score; LEN, lenvatinib; IVR, Interventional Radiology; AFP, α-fetoprotein; DCP, des-γ-carboxy prothrombin.

## Data Availability

Data is contained within the article or supplementary material are available according to “MDPI Research Data Policies” at https://www.mdpi.com/journal/cancers/instructions#suppmaterials.

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
