# Peer review of "Alternating Lenvatinib and Trans-Arterial Therapy Prolongs Overall Survival in Patients with Inter-Mediate Stage HepatoCellular Carcinoma: A Propensity Score Matching Study"

_cancers, 2021, doi:10.3390/cancers13010160_

Round 1

Reviewer 1 Report

I read with great interest this original article “Alternating lenvatinib and trans-arterial therapy prolongs overall survival in patients with intermediate stage hepatocellular carcinoma: A propensity score matching study”. This is a well-written article. My main point in a clarification needed for the group of patients and the precise treatments received.

In addition, I have additional concerns, which need also to be discussed. Please consider the following comments.

Major comments

  1. Please better explain population of study in the abstract (you do not mention lenvatinib in these criteria). From the abstract, it is hard to understand the role of lenvatinib in this study.
  2. I do not get your groups of patients. All patients received lenvatinib, and among those, 41 received AT and 72 no AT. In that case, please explain and clarify why at least63 patients of the no AT group received TACE (refactory patients)? Please describe in the methods exactly in what consisted the no AT group (did they had received TACE earlier and when?). This makes the design hard to understand.
  3. Figure 1 : This waterfall plot was performed before any TACE/HAIC treatment ? This is not consistent with Figure 2. Please specify in figure1 legend.
  4. If patients received TACE/HAIC in the no AT group, did you not look at these variables in the univariate analysis?
  5. Did you look at hemorrhage adverse events ?
  6. Did you look at hepatic effects of HAIC such as biliary complications of cholangitis ?
  7. Please discuss your results compare to Wang X et al (Esmo 2020 abstract 984P).
  8. Please discuss drugs for HAIC compare with Shi et al study (ESMO 2020 abstract 981O).

Author Response

To REVIEWER 1

Thank you very much for your letter regarding our manuscript (cancers-1039492). We appreciate your comments, which have helped us to improve our manuscript. In line with your comments, please find below our point-by-point responses.

Comment 1) Please better explain population of study in the abstract (you do not mention lenvatinib in these criteria). From the abstract, it is hard to understand the role of lenvatinib in this study.

Answer: As you pointed out, we did not explain the population of this study in the abstract. We enrolled 113 intermediate-stage HCC patients treated with lenvatinib (LEN). Patients were classified into the AT (n=41) or non-AT group (n=72) according to the post LEN treatment. The description was added in the revised manuscript (Page 1 , line 37-39).

Comment 2) I do not get your groups of patients. All patients received lenvatinib, and among those, 41 received AT and 72 no AT. In that case, please explain and clarify why at least 63 patients of the no AT group received TACE (refactory patients)? Please describe in the methods exactly in what consisted the no AT group (did they had received TACE earlier and when?). This makes the design hard to understand.

Answer: We appreciate your comment. We apologize for the unclear description. We did not explain the detail of the patient characteristics in Table 1. TACE in Table 1 indicates the pre-TACE condition before treatment with lenvatinib (LEN). In Table 1, we have replaced the word with “TACE condition before LEN treatment (refractory/not eligible)” (Table 1). In the no AT group before LEN treatment, 63 patients were refractory to TACE and 9 patients were not eligible for TACE. We revised these descriptions in the revised manuscript. (Page 2, line-87-88, Page 3, line 94-95, Table 1). Again, we appreciate your valuable comment, which has helped us to improve our manuscript.

Comment 3) Figure 1: This waterfall plot was performed before any TACE/HAIC treatment? This is not consistent with Figure 2. Please specify in figure1 legend.

Answer: We apologize that we did not clearly describe the waterfall plot in Figure 1. The waterfall plot was performed by the initial response to LEN evaluated with modified RECIST. We added these descriptions in Figure 1 legend and the results session in the revised manuscript. (Page 3 , line 101-line 106, line 109).

Comment 4) If patients received TACE/HAIC in the no AT group, did you not look at these variables in the univariate analysis?

Answer: We appreciate your comment. Following your suggestion, we analyzed the factor associated with patients received TACE/HAIC after treated with LEN in the no AT group. There was no significant difference in the survival time between the LEN-TACE/HAIC group and the LEN-another therapies group (p=0.17, Supplement figure 1). Thus, in the no AT group, there was no impact of TACE/HAIC after treatment with LEN on survival. These findings further support our hypothesis that alternating LEN and trans-arterial therapy was important therapeutic strategy for improvement of overall survival.

(Supplement figure 1)

Described in word.

Comment 5) Did you look at hemorrhage adverse events?

Answer: We appreciate your comment. There was no hemorrhage as adverse events in the enrolled patients of this study. As you indicated, a hemorrhage is a serious adverse event. We have added the data for hemorrhage in Table A1. (Page , 11, line 342).

Comment 6) Did you look at hepatic effects of HAIC such as biliary complications of cholangitis?

Answer: We appreciate your comment. Biliary complications of cholangitis were not seen after treated with HAIC in this study. As your pointed out, biliary complications of cholangitis were seen after treated with HAIC in some cases. Thus, we excluded the patients of history recieved choledochojejunostomy in this study. The description was added in the revised manuscript (Page 7, line 178-180).

Comment 7) Please discuss your results compare to Wang X et al (Esmo 2020 abstract 984P).

Answer: Following your suggestion, we compared our study to Wang X et al. “Sorafenib plus Hepatic Arterial Infusion Chemotherapy versus Sorafenib Alone for Advanced Hepatocellular Carcinoma with Major Portal Vein Tumor Thrombosis (Vp3/4): A Randomized Phase II Trial” (1). Although the tumor stage was different between the two studies, Wang W et al. demonstrated that the prognosis of patients treated with sorafenib plus HAIC prolonged overall survival compared treated with sorafenib monotherapy in their study. Thus, our results were in good agreement with the previous study by Wang X et al (1). This comparative description was added in the discussion session of the revised manuscript. (Page 7, line 198- line 199).

Comment 8) Please discuss drugs for HAIC compare with Shi et al study (ESMO 2020 abstract 981O).

Answer: Following your suggestion, we compared our study to Shi et al. “Hepatic arterial infusion chemotherapy (HAIC) with oxaliplatin, fluorouracil, and leucovorin (FOLFOX) versus trans-arterial chemoembolization (TACE) for unresectable hepatocellular carcinoma (HCC): a randomized phase 3 trial (ESMO 2020 abstract 981O). We totally agreed that HAIC is suitable for large unresectable HCC, such as high tumor burden (multinodular type, huge type, and invasive growth type) as Shi et al. reported (2). In our study, we also selected HAIC for high tumor burden HCC. We added these sentences in the revised manuscript (Page 7 and 8, line 200-line 206).

References

(1) X, Wang.; K, Zheng,; G, Cao., et al. Sorafenib plus hepatic arterial infusion chemotherapy versus sorafenib alone for advanced hepatocellular carcinoma with major portal vein tumor thrombosis (Vp3/4): A randomized phase II trial. Annals of Oncology (2020) 31 (suppl_4): S689, doi: https://www.annalsofoncology.org/article/S0923-7534(20)41096-8/fulltext

(2) Ming, Shi.; Q, Li,; M. He.; R. Guo., et al. Hepatic arterial infusion chemotherapy (HAIC) with oxaliplatin, fluorouracil, and leucovorin (FOLFOX) versus transarterial chemoembolization (TACE) for unresectable hepatocellular carcinoma (HCC): A randomised phase III trial. Annals of Oncology (2020) 31 (suppl_4): S629-S644, doi: https://oncologypro.esmo.org/meeting-resources/esmo-virtual-congress-2020/

Reviewer 2 Report

The study by Shimose et al aimed to determine to identify the therapeutic efficacy of alternating LEN and trans-arterial therapy (AT), including TACE and HAIC, in patients with intermediate-stage HCC. Additionally, the authors applied propensity score matching analysis (PSM) to reduce confounding.

TACE is the standard treatment for intermediate-stage HCC but is ineffective for suppressing tumor growth in multiple HCC nodules (≥7) when administered alone. LEN is up for phase 3 clinical trial for HCC.

This study is quite interesting in that the authors show some evidence on how PSM, AT with LEN and trans-arterial therapy improved prognosis compared to LEN monotherapy in patients with intermediate-stage HCC. Moreover, the decision tree analysis revealed that the AT regime may be recommended for intermediate-stage HCC patients below 70 yrs. old who have ALBI grade 1, which could be helpful in clinical settings.

However:

Major concern: 1) No appropriate control was taken in the study.

2) Also the cohort size was too small.

Author Response

To REVIEWER 2

Thank you very much for your letter regarding our manuscript (cancers-1039492). We appreciate your comments, which have helped us to improve our manuscript. In line with your comments, please find below our point-by-point responses.

Comment 1) No appropriate control was taken in the study

Answer: As you pointed out, the non-AT group was heterogeneous and was not an appropriate control group to the AT group. An appropriate control group should be the LEN+TACE/HAIC group; however, most of such patients had post-treatment such as another molecular target agents and it is difficult to collect a sufficient number of appropriate control patients at present. This is a very important issue and we have added this issue as a limitation of this study. (Page 8, line 230-232)

Comment 2) Also, the cohort size was too small

Answer: As you pointed out, the number of patients in this cohort was too small. Thus, a randomized, controlled, and prospective validation study with a larger number of intermediate-stage HCC patients is required to determine the efficacy of AT. This issue was described as a limitation in the revised manuscript. (Page 8, line 233-234)

Round 2

Reviewer 1 Report

Authors have responded to my comments and improved the manuscript; therefore I agree with publication of this manuscript.